# Understanding Quasi-Static and Dynamic Characteristics of Organic Ferroelectric Field Effect Transistors

**DOI:** 10.3390/mi15040467

**Published:** 2024-03-29

**Authors:** Hanjing Ke, Xiaoci Liang, Xiaozhe Yin, Baiquan Liu, Songjia Han, Shijie Jiang, Chuan Liu, Xiaojian She

**Affiliations:** 1School of Electronics and Information Technology, Sun Yat-sen University, Guangzhou 510275, China; kehj@mail2.sysu.edu.cn (H.K.); liangxc5@mail.sysu.edu.cn (X.L.); liubq33@mail.sysu.edu.cn (B.L.); liuchuan5@mail.sysu.edu.cn (C.L.); 2Institute of Chemistry Chinese Academy of Sciences, Beijing 100190, China; yinxiaozhe@iccas.ac.cn; 3College of Electronic Engineering, South China Agricultural University, Guangzhou 510642, China; 4College of Optical Science and Engineering, Zhejiang University, Hangzhou 310027, China; sjjiang@zju.edu.cn

**Keywords:** ferroelectric, compact model, organic field effect transistors, synaptic

## Abstract

Leveraging poly(vinylidene fluoride-trifluoroethylene) [(PVDF-TrFE)] as the dielectric, we fabricated organic ferroelectric field-effect transistors (OFe-FETs). These devices demonstrate quasi-static transfer characteristics that include a hysteresis window alongside transient phenomena that bear resemblance to synaptic plasticity-encapsulating excitatory postsynaptic current (EPSC) as well as both short-term and long-term potentiation (STP/LTP). We also explore and elucidate other aspects such as the subthreshold swing and the hysteresis window under dynamic state by varying the pace of voltage sweeps. In addition, we developed an analytical model that describes the electrical properties of OFe-FETs, which melds an empirical formula for ferroelectric polarization with a compact model. This model agrees well with the experimental data concerning quasi-static transfer characteristics, potentially serving as a quantitative tool to improve the understanding and design of OFe-FETs.

## 1. Introduction

Organic ferroelectric field-effect transistors (OFe-FETs) have attracted more research attention due to their prowess in memory-based operations and the advantages of organic materials. They rely on the ferroelectric layer that can be switched between two stable polarization states. Depending on the polarization orientation, charges are induced at the ferroelectric/semiconductor interface and thus impact the channel current. As a three-terminal device, the channel conductance of Fe-FETs can be modulated via the gate and drain terminals, offering the advantage of concurrently actualizing processing and memory functions. In particular, Fe-FETs enable synapse characteristics like LTP, STP, and EPSC that have been demonstrated based on the voltage-controllable polarization reversal. By modulating the pulse inputs, the ferroelectric polarization can be influenced and the operation of a synapse can be successfully mimicked. Yan et al. demonstrated a P(VDF-TrFE)-based OFe-FET synapse with MoS_2_ as a semiconductor material [1]. Oh et al. achieved 32 levels of conductance states by using a hafnium–zirconium–oxide ferroelectric material [2]. Sun et al. achieved a recognition rate of 98.3% for the MNIST pattern dataset with a 2T-1FeFET design [3]. 

Extensive research efforts have been made to construct ferroelectric transistors from a material engineering perspective, but the compact modeling of device operation is equally important. To date, most of the existing descriptions of the characteristic behavior of ferroelectric transistors are based on inorganic ferroelectrics such as Hf_0.5_Zr_0.5_O_2_ (HZO) and Pb(Zr_0.5_Ti_0.8_)O_3_ (PZT) [4,5,6,7]. Wang et al. explained the HZO-based device using a calibrated ferroelectric (FE) switching model [4]. Pahwa et al. explained the behavior of PZT-based transistors using a negative capacitance model [5]. However, compact models of ferroelectric transistors based on organic ferroelectric (mainly PVDF polymer) and organic semiconductors remain elusive.

Here, we fabricated the OFe-FET based on a P(VDF-TrFE) polymer and the organic semiconductor material dioctylbenzothienobenzothiophene (C_8_-BTBT) with a typical hysteresis window depending on gate sweeping. By modulating the sweeping rate of gate voltage, transfer characteristics under quasi-static and dynamic conditions are demonstrated. We also present the compact model for OFe-FETs by considering the ferroelectric polarization effect within organic ferroelectric materials and the disordered charge transport within organic semiconductors. The compact model generally well describes the experimental results concerning quasi-static characteristics and can capture the core aspects of device performance. Additionally, synaptic behavior under pulse inputs is successfully mimicked with the devices.

## 2. Device Fabrication

Experimental OFe-FET devices with P(VDF-TrFE) (Sigma-Aldrich, Saint Louis, MO, USA) as the ferroelectric materials, C_8_-BTBT as the semiconductors, silicon substrate as the gate electrode and MoO_3_/Au (5 nm/60 nm) as the drain and source electrodes are shown in Figure 1a. P(VDF-TrFE) films were deposited on the heavily p-doped silicon substrate by spin coating using a solution of 50 mg/mL P(VDF-TrFE) 70/30 mol% dissolved in dimethylformamide (DMF), which was followed by drying and annealing at 135 °C for 2 h in N_2_ atmosphere. The C_8_-BTBT layer was then thermally evaporated with a thickness of 40 nm. Finally, MoO_3_ and Au (5/50 nm) were deposited by thermal evaporation to define the source and drain electrodes. The capacitance of the ferroelectric layer was measured by applying the AC voltage (frequency = 1 kHz) on the capacitor using a Precision LCR Meter E4980A (Agilent Technologies Inc., Santa Clara, CA, USA). The transfer characteristics of the OFe-FETs were measured using a semiconductor analyzer (B1500A, Agilent Technologies Inc., Santa Clara, CA, USA).

The microstructure of P(VDF-TrFE) films shows typical randomly distributed needle-like morphology [8] by Atomic Force Microscopy (AFM) in Figure 1b. To further explore the electrical characteristics of the ferroelectric layer, we fabricated the metal-ferroelectric-metal (MFM) sample by using ITO as the bottom electrode, Au as the top electrode and 200 nm P(VDF-TrFE) film as the insulator layer, as shown in Figure 1c. The capacitance measurements were performed on the MFM devices, and the result exhibits strong hysteresis (butterfly shape, Figure 1d), which stems from the polarization reversal in the film as typical ferroelectric properties [9,10].

## 3. Results and Discussion

### 3.1. Quasi-Static Characteristics

The quasi-static transfer characteristics with gate voltage (*V_G_*) that sweeps from +3 to −3 V, +5 to −5 V, +8 to −8 V, +10 to −10 V and then backward are shown in Figure 2, operating in the linear region with a drain voltage (*V_D_* = −1 V). The current–voltage (*I_DS_*-*V_G_*) curves exhibit a typical p-type behavior and hysteresis with a clockwise direction, which is consistent with the accumulation and depletion of hole carriers during the switching of ferroelectric polarization. As a p-type device, the drain current reaches the maximum value, which is represented by the current peaks in Figure 2a when *V_G_* reaches −3 V, −5 V, −8 V and −10 V, respectively. And the enlarged memory window under higher *V_G_* indicates the enhanced polarization of the FE layer and implies the potential for memory application. For memory application, the difference between the threshold voltages (*V_T_*) of OFe-FET under different voltage sweeping direction is defined as a memory window (MW) (see Figure 2b). The device exhibits an on/off ratio reaching 10^4^ and obvious memory windows under different sweeping ranges, even at low voltages. The memory widow is 5.9 V at the *V_G_* = ±10 V sweeping range, which is comparable to the existing research on OFeFETs [11,12,13]. These results suggest that our device can contribute to the realization of OFe-FET memory devices.

We incorporate the expression of ferroelectric polarization into an analytical current-voltage (I-V) model [14,15] for constructing the compact OFe-FET model. There has been a consensus that the ferroelectric polarization of ferroelectric materials is a function of the electric field [16,17]. The polarization below the applied maximum field can be described as:(1)$$P^{+}=P_{s}\tan h\left(\frac{E-E_{c}}{\delta }\right)+\frac{P_{s}}{2}\left(\tan h\left(\frac{E_{max}+E_{c}}{\delta }\right)-\tan h\left(\frac{E_{max}-E_{c}}{\delta }\right)\right)$$
(2)$$P^{-}=P_{s}\tan h\left(\frac{E+E_{c}}{\delta }\right)-\frac{P_{s}}{2}\left(\tan h\left(\frac{E_{max}+E_{c}}{\delta }\right)-\tan h\left(\frac{E_{max}-E_{c}}{\delta }\right)\right)$$
with
(3)$$\delta =2E_{c}/\ln\left[\left(1+\frac{P_{r}}{P_{s}}\right)/\left(1-\frac{P_{r}}{P_{s}}\right)\right]$$
where *P*^−^ or *P^+^* denotes polarization toward negative or positive polarization, *E* is the applied electric field, *P_r_* is the remanent polarization at the zero applied field, *P_s_* is the saturation polarization, *E_c_* is the coercive field at which the polarization changes the sign, and *E_max_* is the maximum applied electric field and *δ* is the calculated constant.

For OFETs, the I-V relations are defined by connecting the above-threshold *I_above_*, subthreshold *I_sub_*, and off regimes. By including a contact voltage drop, the above-threshold current *I_above_* is written as [15]:(4)$$I_{above}=\frac{\frac{W}{L}\mu C_{i}(V_{GS}-V_{T})}{1+\frac{W}{L}\mu C_{i}R_{c}(V_{GS}-V_{T})}\times V_{DS}{\left\{1+{\left[\frac{V_{DS}}{\alpha_{s}(V_{GS}-V_{T})}\right]}^{m}\right\}}^{-\frac{1}{m}}\times \left(1-\lambda V_{DS}\right)$$
where *W* and *L* are the channel width and length, *C_i_* is the insulator capacitor per unit area, *V_T_* is the threshold voltage, *R_c_* is the contact resistance, *λ* is induced to improve output conductance fitting, *α_s_* models the deviation of the saturation drain voltage *V_DS_* from the (*V_GS_*-*V_T_*) point, and *m* controls the abruptness of linear-to-saturation transition. The mobility *µ* is defined by the power-law expression with gate voltage *V_GS_* for the disordered charge transport in organic semiconductors [18,19]:(5)$$\mu =\frac{\mu_{0}}{V_{aa}^{\gamma }}{\left[-(V_{GS}-V_{T})\right]}^{\gamma }$$
where *µ*_0_ is the semiconductor band mobility, and *V_aa_* and *γ* are fitting parameters. Below the threshold, subthreshold current *I_sub_* is described as shown below [19]:(6)$$I_{sub}=I_{0}\exp\left[-\frac{\ln10}{S}\left(V_{GS}-V_{0}\right)\right]$$
where *I*_0_ is the off current, *V*_0_ is the turn-on voltage, and *S* is the subthreshold swing, which can be calculated from the transfer curve as *S* = $dV_{G}/d{log}_{10}I_{DS}$.

The polarization in the ferroelectric layer corresponds to surface charge density (*P*) [20], which leads to a shift of the switch-on voltage *P*/*C_FE_*, where *C_FE_* is the capacitance of the ferroelectric layer and is calculated as $\varepsilon_{0}\varepsilon_{FE}/t_{FE}$, $\varepsilon_{0}$ is the vacuum permittivity, and $\varepsilon_{FE}$ and $t_{FE}$ are the relative permittivity and thickness of the ferroelectric layer. Here, we merge the analytical I-V model with the ferroelectric model by replacing *V_GS_* in Equations (4)–(6) with the effective gate bias *V_eff_* (see Figure 3a), according to Gauss’s law and the voltage balance relationship. Then, the total OFe-FET current model for all the operation regimes is expressed by
(7)$$I_{DS}=\left|\frac{I_{above}}{2}\left\{1-tanh[B\left(V_{eff}-V_{B}\right)]\right\}\right.\left.+\frac{I_{sub}}{2}\left\{1+tanh[B\left(V_{eff}-V_{B}\right)]\right\}+I_{0}\right|$$
where the transition parameter *B* and transition voltage *V_B_* are induced for accurately modeling the transition region [15]. The continuity of the subthreshold regime and the above-threshold regime is realized by the transition function. The flow of the model is shown in Figure 3a. We have constructed this model to describe the electrical behavior of OFeFETs. By connecting the effective gate bias with the ferroelectric polarization, the transfer curves in an OFeFET can be calculated with the *V_T_* shift and typical memory windows. This model took into account both the organic FETs characteristics and the polarization of ferroelectrics, and the connection between different regions (including *I_off_*, *I_sub_*, and *I_above_*), making it more flexible compared to other existing models [4,5,6,7].

To verify the above model, experimental data are fitted using ferroelectric parameter values *P_s_* = 60 mC/m^2^, *P_r_* = 52 mC/m^2^, and *E_c_* = 30 MV/m, and other device parameters (*W/L* = 6, *µ*_0_ = 0.6 cm^2^/Vs, *V_aa_* = 100 V, *γ* = 1.2, *C_i_* = 80 nF/cm^2^, *m* = 4, *α_s_* = 0.5, *λ* = 0.6) were determined from characteristics of transistors with C_8_-BTBT as the semiconductor and SiO_2_ as the insulator. Notice that for the fitting of four data sets, most parameters including ferroelectric and semiconductor parameters which have been listed above were fixed, except that *B*, *V_T_*, *V*_0_, *V_B_* and *S* were slightly adjusted to fit the curves better under various *V_G_* sweeping ranges. The values of subthreshold swing *S*, threshold voltage *V_T_* and the onset voltage *V*_0_ are extracted from the experimental data. Additionally, a practical way to determine the value of *V_B_* is to select the voltage at which the difference between *I_above_* and *I_sub_* is minimal. The general agreement is obtained as shown in Figure 4, indicating the above model applies to OFe-FETs swept at various *V_G_* ranges and also can form good continuity in all operating regimes. The deviation in Figure 4d may be due to the inaccuracy of the simple empirical description of polarization [20] especially below the coercive field *E_c_*. What is more, in the actual measurement, the hole trapping in the channel would usually induce a negative shift of the threshold voltage *V_T_*, whereas the well-fitted results show that this effect is not critical in our devices.

### 3.2. Dynamic Characteristics

The influence of voltage sweeping rate on the dynamic transfer characteristics is investigated. The sweeping rate (*s*) can be calculated as *s* = Δ*V*/*t_s_*, where Δ*V* is the increment of *V_G_*, and *t_s_* is the measurement time for each *V_G_* step (as shown in the inset of Figure 5a). Using *V_G_* with different periods *t_s_*, in Figure 5a, the device memory window (*MW*) increased and the minimum of subthreshold swing (*S_min_*) decreased as the sweeping rate increased. According to the equivalent circuit of the device (Figure 3b), the voltage division relationship and charge conservation equation can be written as:(8)$$V_{G}=V_{FB}+\frac{Q_{S}}{C_{S}}+V_{FE}\;$$
(9)$$Q_{S}=Q_{FE}=P+C_{FE}\cdot V_{FE}$$
where *V_FB_* is the flat band voltage, *Q_S_* and *Q_FE_* are the charges of the semiconductor and FE layer, *V_FE_* is the voltage drop on the ferroelectric (FE) layer, and *C_FE_* is the capacitance of the FE layer. Considering that *V_FE_* varies with the sweep rate under the dynamic sweep, FE charges (*Q_FE_*) cannot be simply approximated as *P*. Thus, we obtained the voltage drop on the FE layer from Formulae (8) and (9) as follows:(10)$$V_{FE}=\frac{C_{S}(V_{G}-V_{FB})}{C_{S}+C_{FE}}-\frac{P}{C_{S}+C_{FE}}$$

According to FE switching dynamics, the switched polarization is related to the duration period of sweeping voltage on FE, and the polarization needs some response time to fully switch when the electric field changes. Therefore, when the sweep rate of gate voltage increases and the duration period time decreases, the switched polarization charge *P* will be decreased [21]. Thus, according to Equation (10), *V_FE_* increases with reduced *P* at the end of the sweep. As a result, the memory window increases with the larger *V_FE_* and presents an exponential decay: $MW=A+Bexp\left(-\frac{t_{s}}{t_{0}}\right)$, where *A* and *B* are constants over time (see Figure 5b). This result is somehow consistent with the rate-dependent memory window investigated via TCAD simulation by Huang et al. [21].

The extracted minimum value of subthreshold swing (*S_min_*) in the reverse sweep is a function of measurement time for each *V_G_* step (*t_s_*), as shown in Figure 5c. In fast sweeping, spontaneous polarization does not respond timely due to its slow switching [22,23]. That is, a larger *t_s_* induces a larger extent of polarization switch, which then leads to a steeper *S*. The dynamic polarization switching under applied electric field *E* is typically described by the Kolmogorov–Avrami–Ishibashi (KAI) model [24], in which the polarization with time can be expressed by the compressed exponential function. We assume *S_min_* follows: $S_{min}={\left[a-bexp\left(-\frac{t_{s}}{t_{0}}\right)\right]}^{-1}+c$, where $t_{0}$ is the switching time constant of the FE dipole at a certain electric field and *a*, *b*, and *c* are constants over time. We find such an empirical expression is in good agreement with our experimental data.

### 3.3. Synaptic Behavior

The variations in the polarization (*P*) of the ferroelectric domains can lead to the synaptic characteristics of OFe-FETs, which has provoked a significant interest in utilizing ferroelectric transistors as synaptic neurons. In general, by applying voltage pulses to the gate of OFe-FETs, the polarization of the ferroelectric films can be gradually altered, resulting in a consistent increase or decrease in the device current. This change strongly depends on the pulse voltage, pulse width, and the interval time between two pulses, reflecting the accumulation dynamics for the neuron implementation. To this end, we design various schemes of pulse inputs to the OFe-FETs by adjusting the pulse amplitude, number, and interval time.

Negative pulses with fixed pulse numbers (*N* = 5) and different amplitudes were first input (the amplitude of these pulses are −3 V, −5 V, −6 V, −8 V, and −10 V, respectively; see Figure 6a). The response current increases from 83 to 542 nA as the pulse amplitude increases from −3 to −10 V, confirming that higher pulse amplitude results in a wider variation of EPSC. This phenomenon can be explained by the model above. With the large input voltage, the *E_max_* in Equations (1) and (2) increases and induces a more complete polarization switching (*P*), which leads to the enhanced conductance. Figure 6b shows the conductance response to pulse inputs with a fixed pulse number (*N* = 100) and different interval times (8, 12 and 20 ms). A shorter interval time resulted in the larger range of variation of the channel conductance (*G*). For example, *G* varies between 51 and 84 nS in response to the pulses with an interval time of 20 ms, and it varies between 83 and 167 nS in response to the pulses with an interval time of 8 ms. The high conductance state under pulse input with short interval time also resulted in a longer time to return to the initial state, which is similar to the forgetting process of brain memory. In terms of pulse number (*N*), pulses with a fixed amplitude (−4 V), a fixed interval time (12 ms) and different pulse number were input (the pulse number of these pulses are 5, 20, 50 and 100). Figure 6c shows that when the pulse number *N* is 5 (or 100), the conductance increased reaches 86 (or 167 nS) and returns to 1 nS (or 37 nS) immediately after the inputs. Based on these results, we applied a pulse sequence with an amplitude of −10 V (*N* = 20) to obtain the essential synaptic function: long-term potentiation (LTP) as shown in Figure 6d. The conductance maintains at 122 nS even after the pulse inputs are removed for a period of time.

By modulating the pulse amplitude, pulse number, and interval time, we were able to obtain the transition from STP to LTP, in which the fast/slow conductance decay is similar to the learning and forgetting processes of the brain [25]. The conductance update and decay with time *G*(*t*) can be fitted with two exponential functions: $G_{p}(t)=G_{p0}\left\{1-exp\left[-{\left(\frac{t}{t_{p}}\right)}^{n_{1}}\right]\right\}+g_{1}$, $G_{d}(t)=G_{d0}\times exp\left[-{\left(\frac{t}{t_{d}}\right)}^{n_{2}}\right]+g_{1}$, where *G_p_* and *G_d_* are the conductance of potentiation and decay, respectively. *G_p_*_0_, *G_d_*_0_, *g*_1_, and *g*_2_ are parameters related to the initial polarization state. Parameters *t_p_* is the constant related to the ability to switch the polarization, while *t_d_* is the decay constant. Furthermore, *n*_1_ and *n*_2_ are stretched exponents, which can be impacted by the pulse inputs. The experimental data can be well fitted as shown with solid lines in Figure 6d.

## 4. Conclusions

In conclusion, we have successfully fabricated the OFe-FETs which exhibited transfer characteristics with bias-sensitive and rate-dependent hysteresis windows. Quasi-static and dynamic characteristics are obtained by modulating the gate voltage sweeping. Moreover, we developed a compact analytical model for quasi-static transfer curves that delineates the electrical characteristics of OFe-FETs, by integrating the characteristic formula for ferroelectric polarization with a compact OFET model. The concordance between the model and a series of experimental data on transfer characteristics has been verified. These results have provided deeper insights into the behavior of OFe-FETs, particularly in regard to the modulation of subthreshold swing and the width of the hysteresis window by varying the sweeping rate of voltage sweeps. Synaptic plasticity-incorporating features such as excitatory postsynaptic current and both short-term and long-term potentiation is also demonstrated in our device. These results may allow a better understanding of OFe-FETs in memory applications, synthetic synapses, and other advanced electronic applications.

## Figures and Tables

**Figure 1 micromachines-15-00467-f001:**
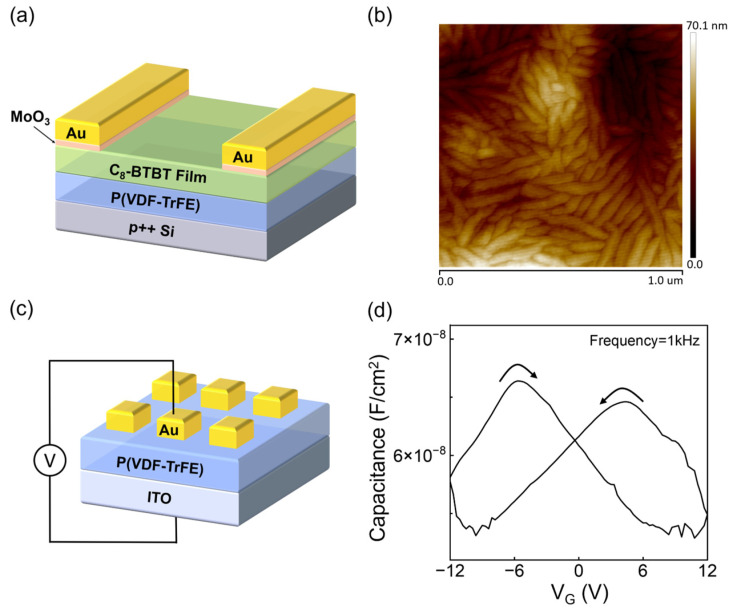
(**a**) Schematic illustration of the OFe-FET structure. (**b**) Typical AFM image of spin-coated P(VDF-TrFE) thin film (scan size: 1 µm × 1 µm). (**c**) Schematic illustration of the MFM structure (the size of the square top electrodes: 200 µm × 200 µm). (**d**) Capacitance versus bias voltage result (swept from −12 to 12 V and then backward) for the MFM capacitor with an insulator layer of ~200 nm P(VDF-TrFE).

**Figure 2 micromachines-15-00467-f002:**
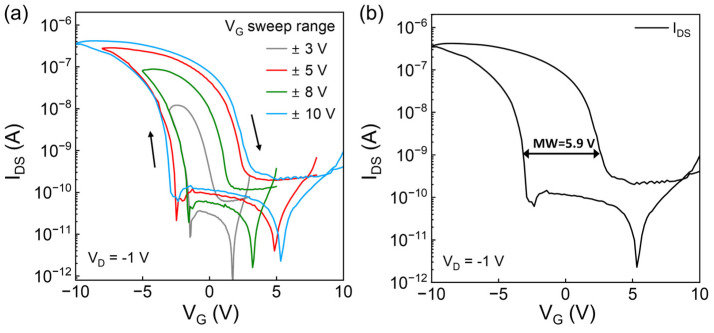
(**a**) Transfer characteristics measured at different *V_G_* sweeping ranges (*V_D_* = −1 V, the sweep direction has been denoted by arrows). (**b**) Schematic of the memory window when *V_G_* sweeps from 10 to −10 V and then backward.

**Figure 3 micromachines-15-00467-f003:**
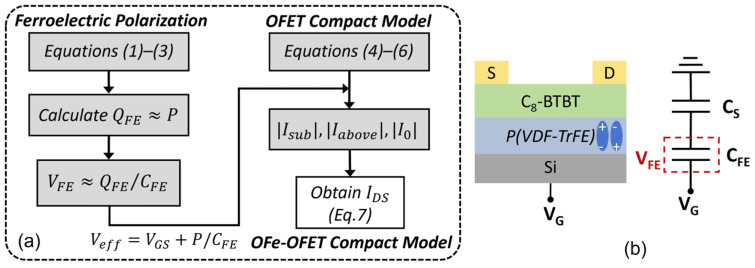
(**a**) Framework of the compact OFe-FET model. (**b**) Equivalent circuit of OFe-FET.

**Figure 4 micromachines-15-00467-f004:**
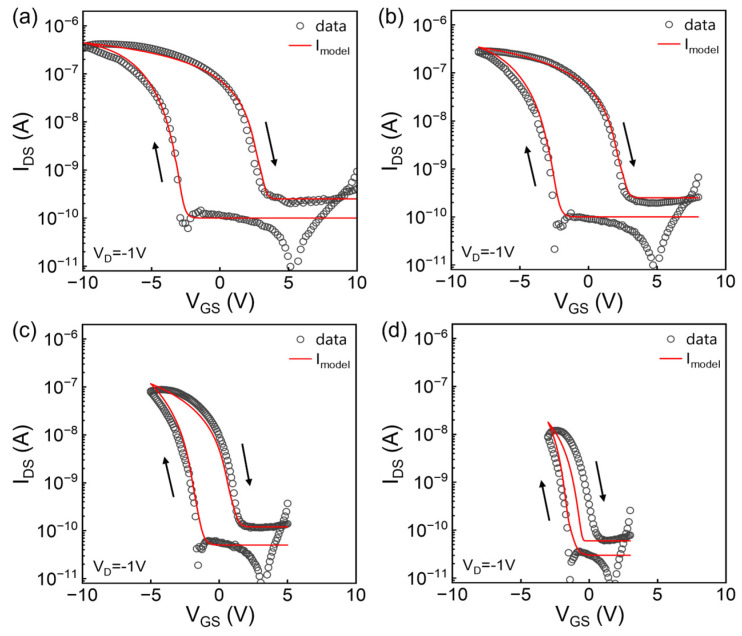
(**a**–**d**) Fitting results of quasi-static transfer curves at different *V_G_* sweeping ranges (symbols represent experimental data and red lines represent the calculated values, the sweep direction has been denoted by arrows).

**Figure 5 micromachines-15-00467-f005:**
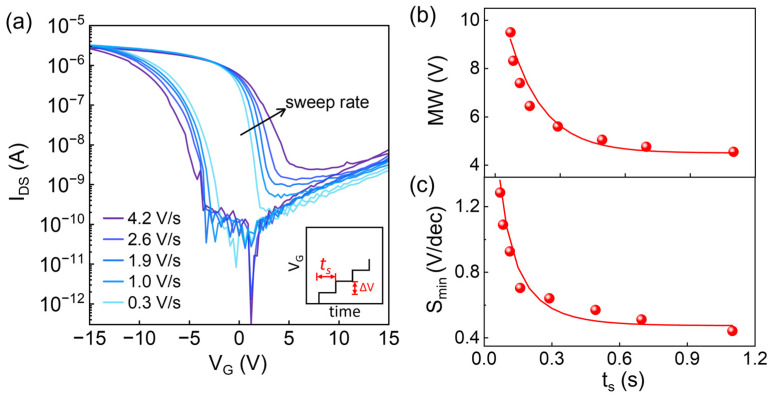
(**a**) Transfer characteristic curves as a function of *V_G_* sweeping rate (inset: *V_G_*-*t* graph under various sweep rate and the definition of *t_s_*). (**b**) Corresponding memory window and (**c**) *S_min_* as a function of *t_s_* (symbols represent experimental results and lines represent the fitting curves with fitting parameters *a* = 1.48, *b* = 1.3, *c* = −0.2, *A* = 4.5, *B* = 7.3, *t*_0_ = 0.16).

**Figure 6 micromachines-15-00467-f006:**
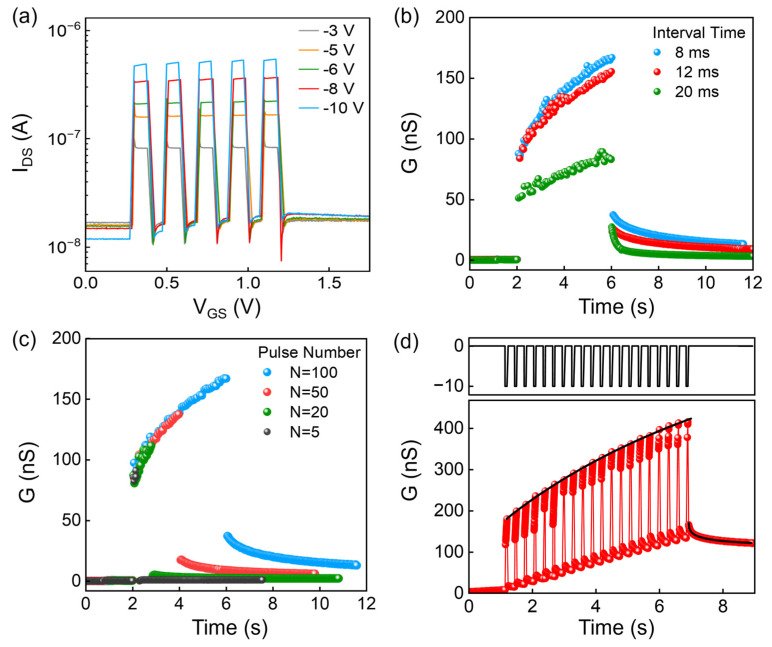
(**a**) Typical EPSC triggered by different pulse amplitude varying from −3 to −10 V. (**b**) Conductance response under varying pulse interval time. (**c**) Conductance response under varying pulse numbers. (**d**) LTP characteristics (−10 V, 60 ms; the red dots and red line represent the experimental data, while the balck solid lines are fitting with exponential functions *G_p_*(*t*) and *G_d_*(*t*)).

## Data Availability

The data that support the findings of this study are available within the article.

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
