# Peer review of "Understanding Quasi-Static and Dynamic Characteristics of Organic Ferroelectric Field Effect Transistors"

_micromachines, 2024, doi:10.3390/mi15040467_

Round 1

Reviewer 1 Report

Comments and Suggestions for Authors

Review Report

Title: Understanding Quasi-static and Dynamic Characteristics of Organic Ferroelectric Field Effect Transistors

Authors: Hanjing Ke, Xiaoci Liang, Xiaozhe Yin, Baiquan Liu, Songjia Han, Shijie Jiang, Xiao-jian She and Chuan Liu.

Summary

This paper presents the fabrication and analysis of organic ferroelectric field-effect transistors (OFe-FETs) using poly(vinylidene fluoride-trifluoroethylene) [(PVDF-TrFE)] as the dielectric. These devices exhibit quasi-static transfer characteristics, including a hysteresis window and dynamic behaviors akin to synaptic plasticity, such as excitatory postsynaptic current (EPSC), short-term and long-term potentiation (STP/LTP). The research introduces an analytical model combining an empirical formula for ferroelectric polarization with a compact transistor model, showing good agreement with experimental data. This model serves as a quantitative tool for understanding and improving OFe-FET design. The study demonstrates the potential of OFe-FETs in memory applications and synthetic synapses, contributing to advanced electronic applications through insights into device behavior, including modulation of subthreshold swing and hysteresis window width by varying voltage sweep rates.

General comments

The paper presents some interesting new results. There is a lack of details in the manuscript that strongly undermines the potential reproducibility of the experiments and leads to a less scientific text.

Major points

·       Some of the figures lack fundamental information and are not fully described:

o   Figure 1a: Is the Silicon substrate intrinsic? (Maybe this information should be added in the main text)

o   Figure 1b: The z-axis legend is not present.

o   Figure 1c: Which is the direction of the sweep?

o   Figure 2: What are the peaks in the IDSvsVG characteristics? Why are they present only in one direction? (Maybe this information should be added in the main text)

o   Figure 2: A graphic identification of the Memory Window could greatly improve the readability of the manuscript.

o   Figure 4: Why are not well-fitted the peaks? Could you put the fitting parameter in the graphs or the main text?

o   Figure 5a: There are too many curves in the graphs. Please use the same number of significant figures on the label. Why at high VGS values are the curves so noisy?

o   Figure 5b: Could be useful to have the time constant on the graph.

o   Figure 5c: Could be useful to have the time constant on the graph.

·       Nowhere in the text is it stated what the gate electrode is.

·       Line 89: Please provide a definition of Memory Window for the readers who are not experts in the field.

·       Line 155-156: Please describe the shape and the characteristics of the signal.

·       Line 207: Please describe the shape and the characteristics of the signal.

Minor points

·       Line 77-81: What is the geometry of such a metal-ferroelectric-metal sample? How did you perform the characterization?

·       Line 138: How are the parameters defined?

·       Line 181: Please define the Smin parameter.

·       Line 189: I think that “constants of time” should be “constant over time”.

Reviewer 2 Report

Comments and Suggestions for Authors

The manuscript reports by using PVDF-TrFE dielectrics to created organic ferroelectric FETs, which exhibit quasi-static transfer characteristics with a hysteresis window and synaptic-like behaviors including EPSC and STP/LTP. By altering voltage sweep speeds, they examined the subthreshold swing and dynamic hysteresis. They developed an analytical model based on ferroelectric polarization, closely matching experimental data, aimed at enhancing OFe-FET design and understanding. The overall information is relatively new and the model would help researchers to better understand organic ferroelectric FETs. I recommend it to be published after addressing the following issues:

1. Could you elaborate on how the presented device performance is compared with other organic ferroelectric FETs? A brief discussion is needed.

2. The proposed analytical model that agrees well with experimental data. Could you discuss how this model compares with some other existing models for ferroelectric FETs?

3. Certain terms, such as CFE, have not been clearly defined or explained within the manuscript. A thorough review is warranted to ensure clarity and comprehensibility throughout the document.

4. The influence of voltage sweep speeds on the subthreshold swing and dynamic hysteresis presents an interesting aspect of the study. It would be insightful if the authors could elaborate on whether these properties significantly impact the synaptic-like behaviors of EPSC, STP, and LTP observed in the devices.

Round 2

Reviewer 1 Report

Comments and Suggestions for Authors

The manuscript has improved greatly in terms of form and content. I congratulate the authors and recommend publication in the current form.